# Room-temperature optically detected magnetic resonance of single defects in hexagonal boron nitride

Hannah L. Stern[1,4 ✉], Qiushi Gu[1,4], John Jarman[1,4], Simone Eizagirre Barker [1], Noah Mendelson[2], Dipankar Chugh[3], Sam Schott[1], Hoe H. Tan [3], Henning Sirringhaus [1], Igor Aharonovich [2] & Mete Atatüre [1 ✉]

Optically addressable solid-state spins are important platforms for quantum technologies, such as repeaters and sensors. Spins in two-dimensional materials offer an advantage, as the reduced dimensionality enables feasible on-chip integration into devices. Here, we report room-temperature optically detected magnetic resonance (ODMR) from single carbon-related defects in hexagonal boron nitride with up to 100 times stronger contrast than the ensemble average. We identify two distinct bunching timescales in the second-order intensity-correlation measurements for ODMR-active defects, but only one for those without an ODMR response. We also observe either positive or negative ODMR signal for each defect. Based on kinematic models, we relate this bipolarity to highly tuneable internal optical rates. Finally, we resolve an ODMR fine structure in the form of an angle-dependent doublet resonance, indicative of weak but finite zero-field splitting. Our results offer a promising route towards realising a room-temperature spin-photon quantum interface in hexagonal boron nitride.

[1] Cavendish Laboratory, University of Cambridge, J.J. Thomson Ave., Cambridge CB3 0HE, UK. [2] ARC Centre of Excellence for Transformative Meta-Optical Systems, Faculty of Science, University of Technology Sydney, Ultimo, NSW, Australia. [3] ARC Centre of Excellence for Transformative Meta-Optical Systems, Research School of Physics and Engineering, The Australian National University, Canberra, ACT, Australia. [4] These authors contributed equally: Hannah L. Stern, Qiushi Gu, John Jarman. ✉email: hs536@cam.ac.uk; ma424@cam.ac.uk

D efects in wide band-gap materials can host optically active confined spins that act as artificial atoms in convenient and scalable platforms[1,2]. Colour centres in diamond[3–5] and silicon carbide[6,7] are prime examples of such systems, with long spin coherence times[8] and high-fidelity spin control and read-out[9]. Their coupling to nuclear spins further enables the realisation of optically accessible long-lived quantum memories[1,10–14]. Together with nanofabrication capabilities, these features make impurity spins leading candidates for light-based quantum information, sensing and communication technologies[4,15–20]. However, out-coupling light from defects in bulk crystals can be challenging and most defects in bulk materials require low-temperature operation.

Layered van der Waals materials are an alternative platform[21–25], where single-photon emitting defects are reported to be among the brightest to date[26] and the reduced dimensionality may allow for a feasible route to designing scalable two-dimensional quantum devices[27,28]. Hexagonal boron nitride (hBN) is a two-dimensional van der Waals crystal that was recently shown to host a plethora of defects that display sharp photoluminescence (PL) spectra at room temperature ranging from 580 nm to 800 nm[26,29,30], which can be tuned spectrally via strain and electric field[31–34]. Multiple defect classes are emerging in hBN: a structure involving a single negatively charged boron vacancy ($VB^-$) displays broad emission at 800 nm and optically detected magnetic resonance (ODMR); however, this defect has only been measured on the ensemble level[35–38]. There are also individually addressable defects around 700 nm, where the presence of spin has been inferred via their magneto-optical signature[39], and recently via cryogenic ODMR measurements in crystalline hBN[40]. A family of narrow-band bright emitters with distinctly sharper zero-phonon lines (ZPL) in the visible spectral range[41–52] has recently received more attention; they can be created controllably via chemical vapour deposition (CVD)[44–47] and plasma treatment methods[48], display spectrally narrow bright optical emission[49], and have already been integrated into optical cavities[50–52]. As such, they hold significant potential towards room-temperature devices for quantum-photonic applications; yet accessing their inherent spin at single-defect level is required for their implementation as a room-temperature spin-photon interface.

In this article, we demonstrate that single defects in hBN host optically addressable spins at room temperature. We investigate hBN with well-isolated single defects that have recently been assigned to carbon impurities and show that they present strong optical signatures of single spins at room temperature. We find that the single-defect ODMR contrast can reach beyond 30%, approximately 100-fold stronger than the 0.4% contrast we observe for the high-density ensemble measurements of the same type of defect[47]. Strikingly, we also observe a bipolar ODMR response across defects and this bipolar nature of the ODMR contrast is explained by our kinetic model. Through second-order intensity-correlation measurements per defect, we further show that the presence of ODMR is correlated strongly with the presence of a second bunching timescale. Finally, below-saturation ODMR lineshape measurements and a spin model simulation reveal that defects exhibit an angle-dependent doublet resonance, consistent with a $S > 1/2$ system with modest zero-field splitting. Our results represent an important milestone for the development of room-temperature quantum optical platforms based on individually accessible qubits in two-dimensional materials.

## Results

**Material characterisation.** To compare the behaviour of single hBN defects with the behaviour of previously reported defect ensembles, we measure a series of multilayer hBN films with varying defect density, where the optical emission has been associated with carbon impurities[47] (see Methods). The material is grown via an MOVPE process that results in hBN layers with a rough surface profile[53] and clear wrinkles that can be seen in confocal images (Supplementary Figs. 1, 2). The films show increasing levels of carbon-boron and carbon-nitrogen bonding which in turn correlates with the defect density and brightness of the material under 532-nm illumination[47].

Figure 1 presents representative optical properties for the defect ensemble in the high-density material (panel a) and for defect A, a typical isolated defect in the low-density material (panels b and c). The insets of the panels a and b include integrated-PL intensity as a function of optical excitation power, as well as integrated-PL confocal images showing the defect density for the two materials. In contrast to the broad PL spectrum for the ensemble (Fig. 1a), the single-defect spectrum in Fig. 1b comprises well-resolved ZPL and multiple phonon sidebands (PSB) with an energy tuning of ~180 meV, consistent with previous reports[47]. Figure 1c is the non-background corrected second-order intensity-correlation measurement ($g^{(2)}(\tau)$) on the integrated-PL intensity for defect A. The antibunching behaviour shows $g^{(2)}(0) = 0.34(3)$ (Fig. 1c inset), indicating that defect A is an isolated single defect (Supplementary Fig. 11 for background-subtraction analysis).

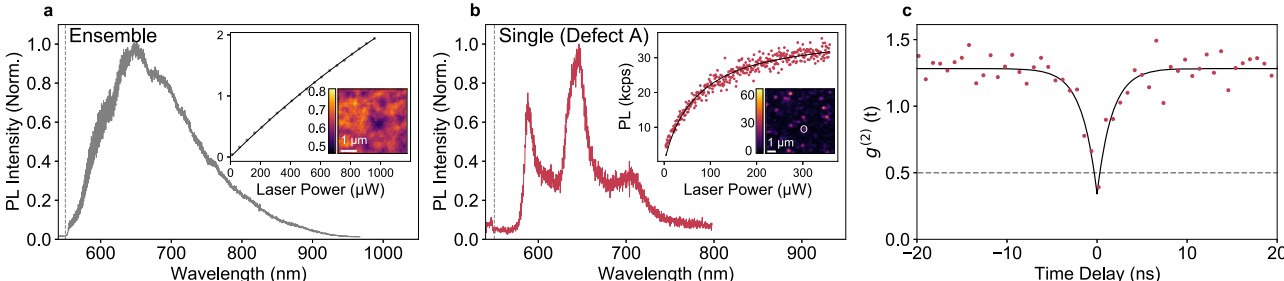

**Fig. 1 Optical properties of single and ensemble hBN defects. a** Normalised PL spectrum of ensemble defects under 532-nm laser excitation. Dashed vertical line represents the cut-off of a 550-nm long-pass filter. Inset: 5 × 5-μm image map of the integrated-PL intensity and example laser-power dependence of PL counts at a fixed location. The colour bar scale is in units of million counts per second (Mcps). **b** Normalised PL spectrum of single defect A under 532-nm laser excitation. Dashed vertical line signifies the 550-nm long-pass filter. Inset: 10 × 10-μm image map of the integrated-PL intensity with a white circle around defect A, and the power dependence for defect A, showing optical saturation power ($P_{sat}^{optical}$) = 70(2) μW and saturated PL intensity ($I_{sat}$) = 37,900(300) counts/s. $P_{sat}^{optical}$ reflects the power needed to achieve half the saturated PL counts (see Supplementary Fig. 3). The colour bar scale is in units of thousand counts per second (kcps). **c** Second-order intensity-correlation measurement for defect A at $1.4P_{sat}^{optical}$ excitation, solid black curve is a theoretical fit. The fitted $g^{(2)}(0) = 0.34(3)$ and optical lifetime of 1.60(1) ns. Background analysis in Supplementary Fig. 11.

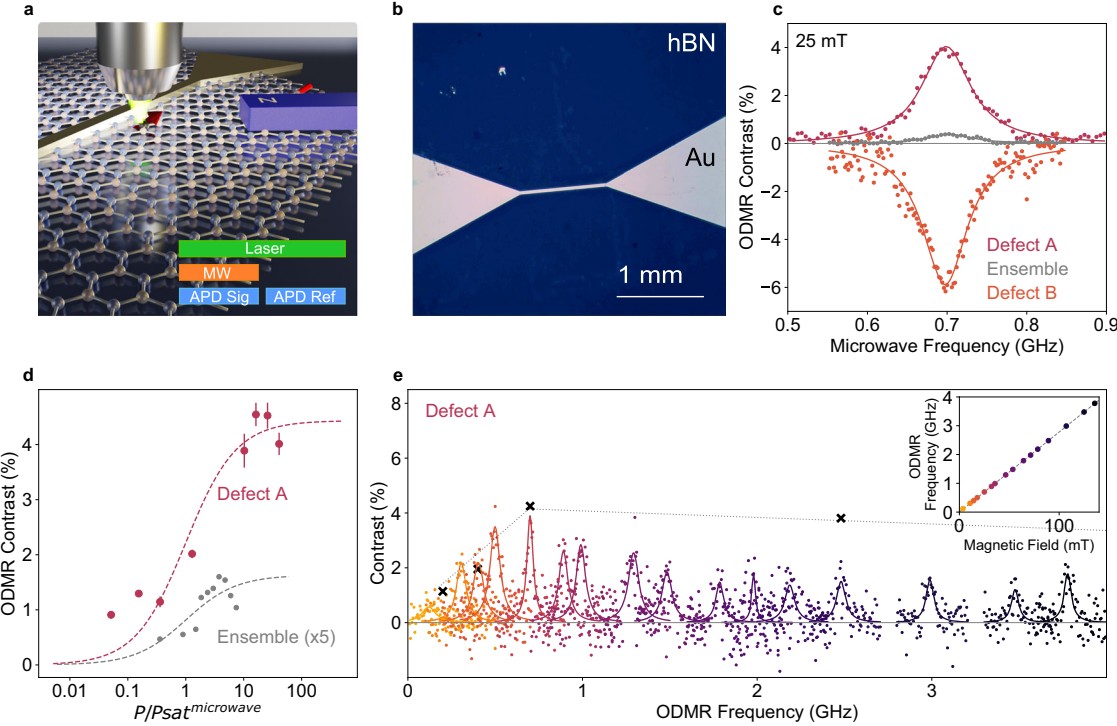

**Fig. 2 Room-Temperature ODMR Setup and Saturated ODMR Measurements. a** An illustration of the measurement setup showing a permanent fixed magnet positioned in-plane relative to the hBN layers with lithographically patterned microstrip. Inset schematic representing one cycle of the ODMR protocol: a microwave pulse (orange) for the first half of the lock-in cycle, and signal and reference counts that are measured by the single-photon counting detectors (APD) (blue). The excitation laser is present for the full lock-in cycle (green). **b** An optical image of a lithographically patterned 1-mm-long gold microstrip used to apply microwave field to the hBN defects. **c** Room-temperature saturated ODMR of single defects (defect A, red circles; defect B, orange circles) and of the ensemble (grey circles), all measured at 25-mT external in-plane magnetic field and at 10 times the microwave saturation power ($10P_{sat}^{microwave}$). $P_{sat}^{microwave}$ refers to the microwave power needed to achieve half the saturated ODMR contrast for the given defect or ensemble. The solid curves are Lorentzian fits to the ODMR lineshapes. We determine a saturated linewidth of 34(3) MHz, 37(2) MHz and 34(2) MHz for the ensemble, defect A and defect B, respectively. **d** ODMR contrast as a function of normalised microwave power ($P/P_{sat}^{microwave}$) for defect A (red) and the ensemble (grey) with error bars showing one standard deviation. **e** ODMR spectra for defects A as a function of in-plane magnetic field (each colour represents a different magnetic field strength), measured at 0.2 W, which is $10P_{sat}^{microwave}$ at 25 mT and $2P_{sat}^{microwave}$ at 89 mT. A constant microwave power was used across the magnetic field range as higher powers cause microwave-induced heating. The black crosses, linked by a dashed line, mark the saturated contrast at that magnetic field strength, i.e., 1.1% and 1.9% at 7 mT and 14 mT, respectively, saturating at ~4% at 25 mT and beyond (Supplementary Fig. 27). The inset shows the ODMR resonance central frequency for each measurement shown in (**e**) (measurements plotted in the same colour) against the magnetic field strength the measurement was performed with, fit to a linear function with a g-factor of 1.98(3).

**Optically Detected Magnetic Resonance.** Figure 2a illustrates the basic elements of our continuous-wave ODMR setup. We record integrated-PL intensity under 532-nm laser excitation as a function of the applied microwave field. We modulate the amplitude of the microwave field with a square wave at 70 Hz to determine the difference between the PL intensity when the microwave field is applied (signal) and when the field is not present (reference). The difference in PL is normalised by the reference PL intensity to obtain an ODMR contrast for each microwave frequency. This eliminates contributions from slow variations during each measurement. A permanent magnet mounted on a linear translation stage tunes the amplitude of the external magnetic field at the defect, which is applied in-plane relative to the hBN for the first measurements presented. Figure 2b shows an optical image of one of our hBN devices, showing the lithographically patterned microstrip on the hBN layer used to deliver the microwave field locally. The microstrip is deposited on top of the grown hBN multilayers, which uniformly span the image.

Figure 2c presents example ODMR spectra for the ensemble (grey circles) and two single defects (red and orange circles) with a 25-mT in-plane magnetic field, using a microwave field high enough to saturate the ODMR contrast. All three saturated ODMR signals are at 700-MHz central frequency and show a

~35-MHz linewidth. Strikingly, the single-defect ODMR signal has substantially higher contrast with respect to that of the ensemble, up to 100-fold for some defects (Supplementary Table 3). The comparable linewidth observed for the ODMR spectrum of the high-density ensemble, and the single defects suggest that the mismatch might arise from a possibly low fraction of spin-active defects, similar to previous reports[39], as opposed to other effects such as spectral broadening of the ODMR resonance. Indeed, out of more than 400 isolated defects we investigated for this work, 27 revealed measurable ODMR signal with fixed external magnetic field strength and orientation, suggesting a yield in our experiments of ~5%. Further, ODMR signals of different signs are measured across different defects: defects A and B in Fig. 2c are presented as examples of the positive and negative ODMR contrast that we observe across the ODMR-active defects, with a roughly even yield of each polarity (Supplementary Table 3). A positive (negative) ODMR signal indicates that microwave drive at spin resonance frequency leads to an increased (decreased) PL intensity, which can further contribute to the modest ODMR signal from the ensemble. Figure 2d presents the ODMR contrast of defect A as a function of microwave power at 25-mT applied magnetic field, demonstrating the expected saturation behaviour. The ensemble ODMR

contrast shows equivalent saturation behaviour albeit at a significantly lower ODMR signal.

An ODMR frequency of 700 MHz at 25 mT is consistent with a g-factor of ~2, typical for atomic spin defects in solids and Fig. 2e presents the evolution of the ODMR spectra for defect A. The ODMR spectra in Fig. 2e are all acquired at a fixed input microwave power ($10P_{sat}^{microwave}$ at 25 mT), to compromise between microwave-induced heating at high microwave field and ODMR signal strength at low microwave field. The apparent variation of contrast, common to all defects, is due to the frequency-dependent microwave transmission into the micro-strip. The black crosses and dashed line highlight the saturated ODMR contrast for the corresponding spectra, which shows that the maximum ODMR response for defect A builds up to a steady contrast of ~4% as a function of the magnetic field (Supplementary Fig. 19). The inset presents the magnetic-field-dependent shift of the central frequency for the ODMR signal for these defects. A linear fit to the plot reveals a g-factor of 1.98(3) in line with the g-factor measured for other defects and with 2.03(3) measured for the ensemble (Supplementary Fig. 28).

**Bunching dynamics and the observation of ODMR.** Second-order intensity-correlation ($g^2(\tau)$) measurements were performed out to 1 ms time delays on the hBN defects to analyse the timescales associated with the optical transitions. In Fig. 3, $g^2(\tau)$ measurements are shown for defect B (panel a) and a second defect that did not show ODMR (panel b). For both defects, the $g^2(\tau)$ data is fit to bi-exponential and tri-exponential decay functions (Eqs. (1) and (2)) (tri-exponential not shown in (b)), which allows us to determine the antibunching ($\tau_{ab}$) and bunching ($\tau_b + \tau_{b(additional)}$) time-scales (panels d–f). We apply the same analysis across 18 defects

(Supplementary Figs. 4–10), half of which show ODMR, and we find a wide range in bunching timescales, consistent with the previous reports[26,39,43]. However, interestingly we observe a strong correlation between the presence of ODMR and the presence of two bunching timescales, independent of the laser power we use (Supplementary Fig. 12). This is shown in panel c, where tri-exponential fits show that one of the two bunching timescales (denoted $\tau_{b(additional)}$) for the ODMR-inactive defects shows a significant error $\left(\frac{\sigma_{b(additional)}}{\tau_{b(additional)}}\right)$ associated with the fit. This indicates that while the photodynamics of ODMR-active defects is best described with two bunching timescales, non-ODMR defects display only one. We find that the additional bunching timescale ($\tau_{b(additional)}$) ranges from 90 ns to 5.3 µs for the ODMR-active defects, which is shorter than the other bunching timescale ($\tau_b$), which ranges from 10 to 350 µs for all defects.

$$g^{(2)}(\tau) = y_0 - ae^{\left(-|\tau-t_0|/\tau_{ab}\right)} + be^{\left(-|\tau-t_0|/\tau_b\right)} \quad (1)$$

$$g^{(2)}(\tau) = y_0 - ae^{\left(-|\tau-t_0|/\tau_{ab}\right)} + be^{\left(-|\tau-t_0|/\tau_b\right)} + ce^{\left(-|\tau-t_0|/\tau_{b(additional)}\right)} \quad (2)$$

A simple three-level model with ground-state spin captures the correlation between ODMR and two bunching timescales. In this model the appearance of two bunching timescales arises from an imbalance in shelving and de-shelving rates between the spin sublevels of the optical manifold and the metastable state

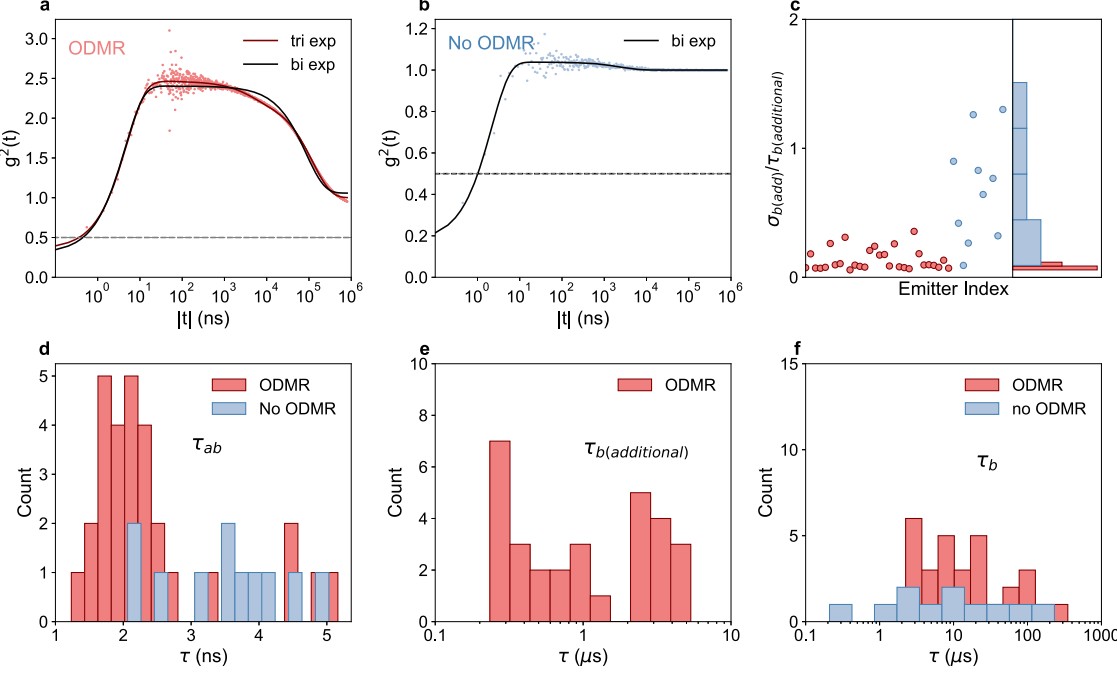

**Fig. 3 Bunching timescales of hBN ODMR-active defects. a** Second-order intensity-correlation ($g^2(\tau)$) measurement of an ODMR-active defect, defect B, measured at $1.5P_{sat}^{optical}$ excitation (laser-power saturation in Supplementary Fig. 3), showing the dynamics out to 1-ms time delay ($t = \tau - t_0$), non-background corrected. The data (circles) is fit to a bi- and tri-exponential fit (solid curves). **b** $g^2(\tau)$ measurement of a defect that does not show ODMR, out to 1-ms delay, measured at $0.2P_{sat}^{optical}$ excitation, non-background corrected. The grey circles are the data and the solid line is a bi-exponential fit. For background correction analysis see Supplementary Fig. 11. **d–f** The distribution of antibunching ($\tau_{ab}$) and bunching ($\tau_{b(additional)}$ and $\tau_b$) timescales from 40 measurements of 18 defects. Data for defects that show ODMR is in red and defects that do not show ODMR in blue. Defects that don't show ODMR are not plotted in (**e**) because this data contains high error, as shown in (**c**). **c** A scatter plot (left plot) and histogram (right plot) of the error on the fractional error on the fit $\left(\frac{\sigma_{b(additional)}}{\tau_{b(additional)}}\right)$ of the additional bunching timescale, for ODMR and non-ODMR-active defects.

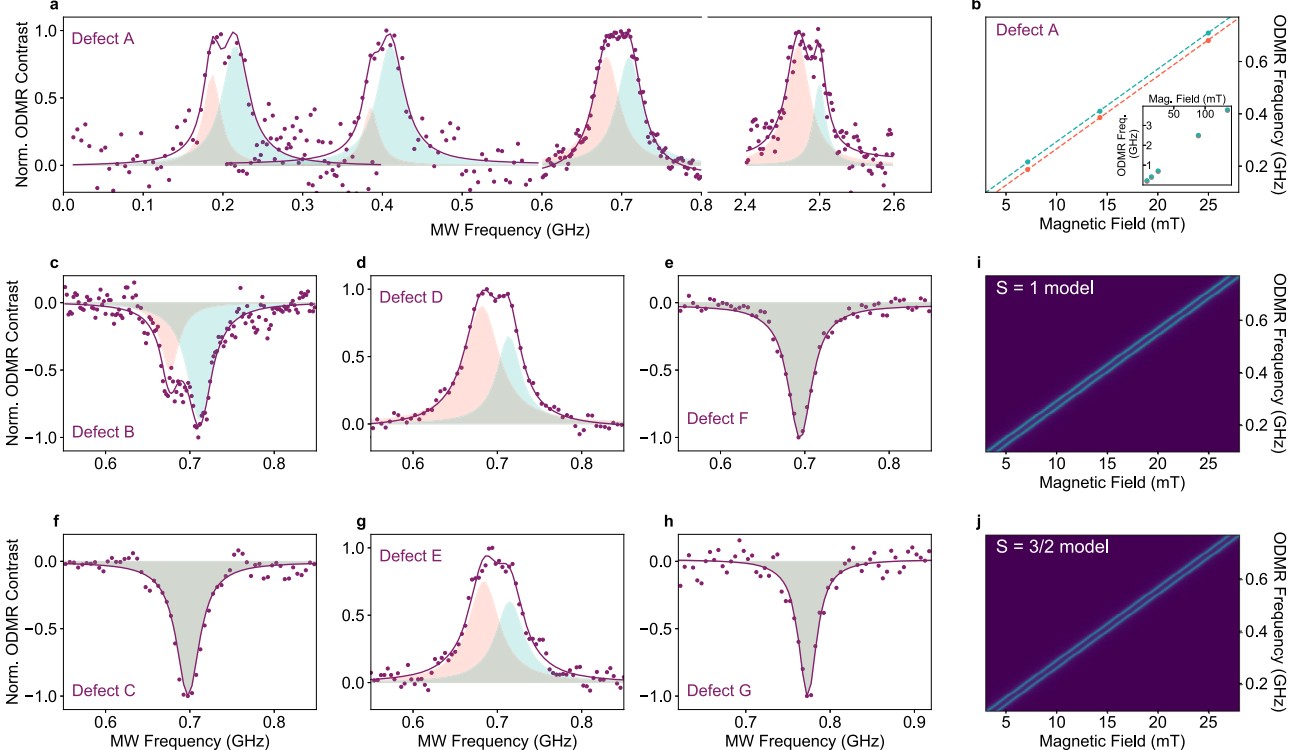

**Fig. 4 Sub-linewidth structure and angle-dependence of below-saturation ODMR spectra. a** Unsaturated ($P_{sat}^{microwave}$ and $1.2P_{sat}^{optical}$) normalised ODMR spectra for defect A with double-Lorentzian fits at a range of magnetic-field strengths. Coloured circles are the experimental results obtained at 7 mT, 14 mT, 25 mT and 89 mT, coloured solid curves are double-Lorentzian fits. For each panel, the shaded blue and red Lorentzian lineshapes show the two components of the doublet. **b** The ODMR frequency of each component of the double-Lorentzian fit, as a function of magnetic field. **c–h** Below-saturation normalised ODMR spectra for defects B, C, D, E, F and G obtained at 25 mT with a fixed in-plane magnet position. Shaded regions indicate the two components of the doublet Lorentzian fits. (**i**) ODMR simulation for $S = 1$ system with $D = 14$ MHz and $E = 4$ MHz (**j**) ODMR simulation for $S = \frac{3}{2}$ system with $D = 7$ MHz and $E = 2$ MHz. In both (**i**) and (**j**) the magnetic field applied along the lab frame axis $z$ and the defects symmetry axis (principal axis of defect's **D** tensor) $z'$. The signal intensity is normalised to 1 and represented by the blue shaded region.

(for detailed model see Supplementary Figs. 13–18). This rate imbalance leads to ground-state spin polarisation, associated generation of ODMR contrast, and the appearance of two bunching timescales. However, if all spin sublevels couple equally to the metastable state, the defects will show only one bunching timescale but no spin polarisation. Our model uses the same three-level structure as other reports[40,54], but it should be noted that it also requires the addition of laser-power-dependent shelving and de-shelving rates.

**ODMR fine structure of hBN defects**. To resolve sub-linewidth features in the ODMR spectrum, we operate at low microwave driving conditions to avoid power broadening. As such, we operate in the near-optimal regime of signal strength with microwave excitation power at $P_{sat}^{microwave}$, as inferred from saturation measurements (Supplementary Fig. 19, 20). Figure 4 presents the corresponding below-saturation ODMR spectra obtained with the external magnetic field applied in the plane of the hBN layers, for the defects labelled A to G. The coloured circles are the data, and the solid curves are the Lorentzian fit (Supplementary Figs. 21–25 for Gaussian and Voigt fit analyses). For most of the defects (~80%), we resolve a doublet structure, while for others we cannot resolve a splitting. The corresponding panels in Fig. 4 show the constituent individual lineshapes of the doublet resonances (shaded red and blue), obtained with a double-Lorentzian fit.

For the defects where we resolve doublets, the splitting is independent of the magnetic-field strength. Figure 4a demonstrates such independence of the doublet splitting from the in-

plane magnetic-field strength for defect A and panel b displays the central peak frequency for each Lorentzian of the doublet. The splitting for defect A is ~30 MHz across the magnetic field range from 7 mT to 89 mT, and the average linewidth of the constituent single resonances is ~20 MHz. However, the measured doublet splitting varies across defects, between 19 and 50 MHz with a mean splitting of 34(8) MHz (Supplementary Table 3). This continuum of values suggests that the observation of singlet resonances in some defects could be due to the presence of a doublet with a splitting too small for us to resolve.

In principle, both crystal-field in the high-field regime and hyperfine coupling can lead to a split doublet in the ODMR spectrum. Electron paramagnetic resonance (EPR) measurements have shown that electronic spins in hBN couple to nitrogen, carbon and boron nuclear spins[35,55,56]. However, the predicted hyperfine constants and the corresponding splitting for boron[55] isotopes differ starkly from our results. Potential single-carbon substitution defects, ($C_N$ and $C_B$) are predicted to show broadened resonances, rather than a distinct 30-MHz splitting[40,57]. Coupling to one $C^{13}$ nuclei could in principle result in a doublet, however the abundance of $C^{13}$ (~1%) does not reconcile with the yield of ODMR-active defects we measure (~5%)[57]. The hyperfine constant for nitrogen is in the correct range[35], but we do not expect a doublet spectrum from electron-nitrogen coupling. All these make it difficult to assign the ODMR doublet to hyperfine coupling without considering a more complicated atomistic structure.

An alternative origin for the ODMR doublet is zero-field splitting of a $S > \frac{1}{2}$ state. To explore this possibility, we simulate

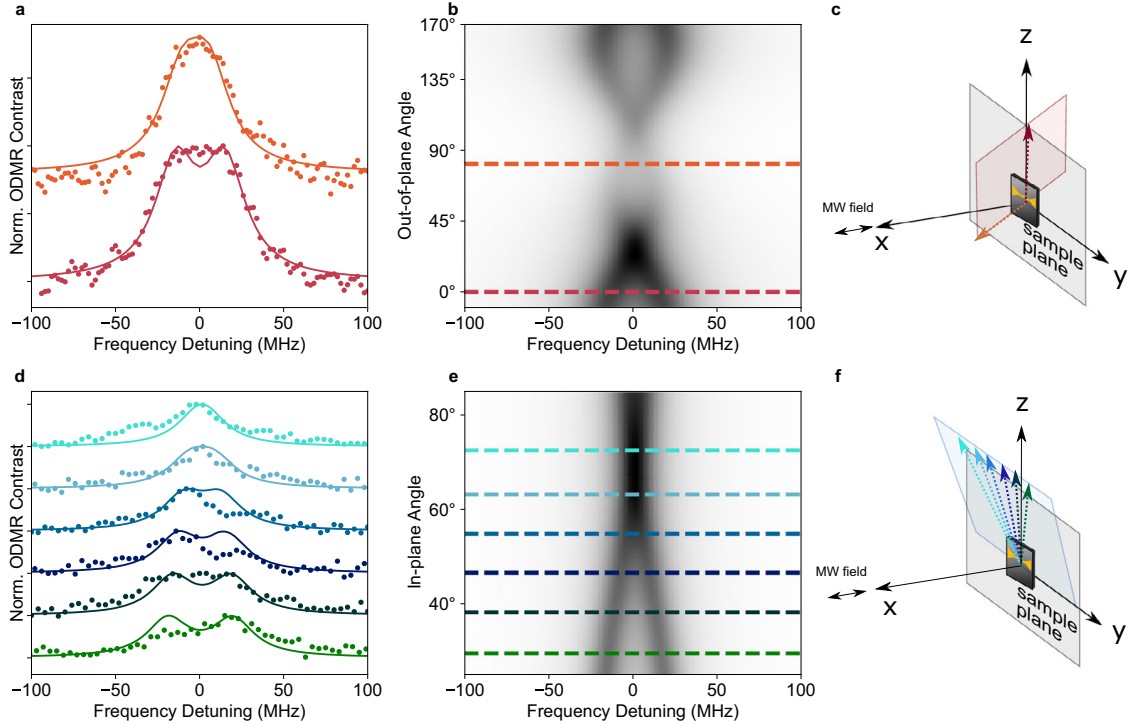

**Fig. 5 Angular dependence of ODMR below saturation. a** ODMR spectra (circles) for defect A at two magnetic field orientations, along the $z$-axis (red) and in the $x$–$y$ plane (orange), magnetic field vectors represented in (**c**). The solid curves are fits to the data using a $S = 1$ model with $D = 25$ MHz, $E = 5$ MHz and the defect **D** tensor symmetry axis ($z'$) rotated in the plane of hBN. **b** The simulated ODMR contrast for defect A through the red plane shown in panel (**c**). The shaded area represents signal intensity normalised to 1. **c** Schematic showing the magnetic field orientations (red and orange vectors) for measurements in (**a**) and (**b**). The red vector points along $z$, and the orange points out of the $xy$ plane. **d** ODMR spectra (circles) for defect E at a series of magnetic field orientations through a plane, shown in panel (**f**). The solid curves are the fit the $S = 1$ model with $D = 25$ MHz, $E = 5$ MHz and $z'$ rotated in the hBN plane (**e**) The simulated ODMR contrast for defect E through the blue plane shown in panel (**f**). **f** Schematic showing the magnetic field orientations (blue vectors) for measurements in (**d**) and (**e**). The blue vectors point at different angles in $zy$ plane with a projection onto $x$. The sample orientation is shown in both (**c**) and (**f**) but the cartoon image (in $zy$ plane).

the expected ODMR spectra for both $S = 1$ and $S = \frac{3}{2}$ systems, using low zero-field splitting parameters and an in-plane magnetic field applied down the principal symmetry axis of the defect **D** tensor (panels i and j of Fig. 4). Neglecting hyperfine coupling, the spin state for a given defect can be described by a spin Hamiltonian in the form,

$$H = g\mu_B \mathbf{B} \cdot \mathbf{S} + D\left(S_z^2 - \frac{1}{3}S(S+1)\right) + E\left(S_x^2 - S_y^2\right), \quad (3)$$

where $\mu_B$ is the Bohr magneton, **S** is the spin projection operator, $g$ is the g-factor, **B** is the external magnetic field, and $D$ and $E$ are the zero-field splitting parameters. For $S = 1$ the two ODMR transitions correspond to transitions between the $m_s = 0$ and the $m_s = \pm 1$ states and are separated by an energy of $2D$. For a $S = \frac{3}{2}$ system, the spin transitions that result in ODMR contrast are $m_s = -\frac{3}{2}$ to $m_s = -\frac{1}{2}$ and $m_s = +\frac{1}{2}$ to $m_s = +\frac{3}{2}$, split by $4D$ in energy[58]. As we can see in panels i and j, using slightly different $D$ and $E$ values, both spin multiplicities can produce a doublet that does not change with magnetic field strength (for more details see SI, section 5).

The appearance of both doublets and singlets in the ODMR measurements can also be explained by a $S > \frac{1}{2}$ model, if we consider that not all defects will have the same orientation in the hBN sample. For both $S = 1$ and $S = \frac{3}{2}$ situations with low zero-field splitting, the splitting of the ODMR doublet is dependent on the orientation of the external magnetic field relative to the symmetry axis of the **D** tensor (determined by

the dominating eigenvalue of the **D** tensor and denoted $z'$). When $z'$ is aligned with the applied magnetic field, the splitting is determined by $2D$, but if the defect is oriented with the symmetry axis offset from the magnetic field, the splitting is defined by $D$ and $E$. In this case, with $E = 5$ MHz, this results in a splitting that cannot be resolved and instead appears as a single peak. Figure 5 shows results for two defects (panels a and d) where the orientation of the external magnetic field is moved through a series of calibrated orientations relative to hBN plane (vector of magnetic field shown in panels c and f). For these two defects, and all other defects measured (Supplementary Figs. 32–41), the ODMR lineshape shows a splitting that can be tuned with angle and that is well described by the $S = 1$ model with $D = 25$ MHz and $E = 5$ MHz (simulation Fig. 5b, e). The largest splitting we measured, 50 MHz, corresponds to a defect with $z'$ along $z$ (Supplementary Fig. 40). The only free parameter in the model is the orientation of $z'$. We also note that for a small fraction of defects, where a doublet splitting was not resolved, there was no noticeable dependence of the ODMR lineshape on the magnetic field orientation (3 defects we measured). Experimental results for these defects are consistent with defects where $z'$ is tilted out of the 2D plane of the hBN sheets (Supplementary Figs. 37–39).

## Discussion
For the ODMR-active defects, we observe some variation of the contrast and lineshape, but the overall behaviour is remarkably similar across the single defects studied with roughly equal

likelihood of finding positive or negative ODMR contrast sign. This bipolarity is unlike the defects in diamond that show a consistent ODMR sign for a given optical defect whether probed as a single or on the ensemble level. However, our analysis reveals a mechanism where all the ODMR behaviour we observe can be explained by a single type of optically active spin defect presenting highly tuneable photodynamics. Our kinetic analysis shows that hBN defects display a wide range in bunching behaviour, but that the presence of two bunching timescales strongly correlates with the presence of ODMR. In addition, we demonstrate that the ODMR contrast and sign can be determined by the intricate balance of the rates of the shelving and the de-shelving optical transitions for every defect. The exceptional variability of hBN optical rates, perhaps via strain[36], may reflect the tunability of the defect energy levels in this 2D system if it can be controlled. This could open routes to a room-temperature spin-photon interface where the spin readout can be reversibly and easily tuned for use in sensing and memory-assisted quantum networks.

Regarding the spin multiplicity and implications for the chemical structure of the defect, the yield of ODMR (~5%) and prevalence of a doublet in our ODMR measurements (80%) is difficult to reconcile with a $S = \frac{1}{2}$ system with a 1% abundance of $^{13}C$, although not impossible. Instead, we find that spin models with $S > \frac{1}{2}$ and low zero-field splitting parameters are viable alternatives. We find that all the ODMR data for our defects is consistent with a $S = 1$ model with $D = 25$ MHz and $E = 5$ MHz, by tuning the defect symmetry orientation in and out of the hBN plane. We consider that a range of defect orientations is highly likely in this material, where the confocal scans and previous high resolution TEM images of the same material show large regions where the hBN layers are tilted relative to the substrate[53]. An alternative explanation could be that we are measuring a range of different optical defects with low, but variable, $D$ and $E$ parameters, dictated by local strain for example. While this is possible, our analysis indicates that invoking different $D$ and $E$ parameters is not necessary to model the data. Finally, while we demonstrate a spin triplet model is consistent with our data, a $S = \frac{3}{2}$ model is difficult to distinguish from $S = 1$ in this field range and thus cannot be ruled out.

Experimental and theoretical reports indicate that the structure of the defect is likely to contain carbon[40,47,59–63]. Defects in single crystalline hBN that show ODMR only under cryogenic conditions have been assigned to a spin-$\frac{1}{2}$ carbon substitution defect ($C_B$)[40,56]. This defect emits at 730 nm and the ODMR shows a broad 40 MHz resonance, attributed to unresolved hyperfine coupling to neighbouring $^{11}B$ and $^{14}N$ nuclei[56]. It is difficult for us to conclude whether we are measuring the same defect as these reports. While our ODMR shares some features with those in ref. [40], such as defects that show positive and negative contrast, there are also interesting differences: in addition to the difference in ZPL energy, the defects we study show ODMR at room temperature and the majority show a doublet, while the defects in ref. [40] do not show ODMR at room-temperature and no splitting was observed. Most recently, a carbon-trimer structure has become a strong candidate for the single-photon emitting defects with ZPLs ~2 eV in hBN[62]. These defects are thermodynamically likely to be formed[63] and have been modelled to show ODMR with highly variable contrast magnitude and sign, regulated by internal optical rates[54], which is similar to our observations. Therefore, we consider these structures, as well as larger carbon clusters predicted to be $S = 1$[63], as strong candidates for the defect we measure. We note that the magnitude of zero-field splitting indicated by our data (<25 MHz) is small compared to spin defects in diamond and SiC[5,64] and organic molecules[65]. However, we also note that 25 MHz corresponds to a spin-spin

magnetic dipolar coupling parameter of electrons separated by 1.3 nm, which corresponds to 10 bond lengths in hBN. This small dipolar coupling, combined with poorly resolved hyperfine coupling may be consistent with a carbon cluster defect and warrants further investigations to shed light on the atomistic structure.

In conclusion, we report optically accessible spin defects in hBN layers via ODMR measurements at room-temperature. We observe ODMR contrast for single well-isolated defects. The sub-unity yield of the ODMR-displaying defects, as well as the polarity of the ODMR sign, are likely reasons for the significantly reduced ODMR contrast reported previously for an ensemble. We identify an important indicator of ODMR, the presence of two bunching timescales in the $g2(\tau)$ measurements which supports the idea that our variations in ODMR sign and contrast strength could be due to variations in photodynamics across defects, potentially caused by variations in strain. ODMR-active defects possess a double peaked resonance with an average splitting of 35 MHz, consistent with a $S = 1$ state with a zero-field splitting on the order of 25 MHz. Angular-dependent measurements and simulations suggest that this continuum of values arises from a variation in the orientation of defects in the hBN plane. Further experimental and theoretical work will be required to develop a deeper insight into the microscopic structure and photophysics of these defects. Regardless, these results reveal the potential for these defects as a tuneable room-temperature spin-photon interface in a two-dimensional material platform.

## Methods

**HBN Sheets**. hBN was grown by metal organic vapour phase epitaxy (MOVPE) on sapphire, as described in Mendelson et al[47]. Briefly, triethyl boron (TEB) and ammonia were used as boron and nitrogen sources with hydrogen used as a carrier gas. Growth was performed at low pressure (85 mBar) and at a temperature of 1350 °C. Isolated defects and ensemble defects were generated by modifying the flow rate of TEB during growth, a parameter known to control the incorporation of carbon within the resulting hBN sheets. For PL measurements, hBN sheets were transferred to SiO₂/Si substrates, using a water-assisted self-delamination process to avoid polymer contamination. Before measurements each device was treated in a UV/ozone cleaner for 15 min.

**Confocal microscopy**. Optical measurements were carried out at room temperature under ambient conditions using a home-built confocal microscopy setup. A continuous-wave 532-nm laser (Ventus 532, Laser Quantum) was sent through a 532 nm band-pass filter and focused on the device using an objective lens with 100X magnification and a numerical aperture of 0.9. Control over excitation power was provided by an acousto-optic modulator (AA Optoelectronics), with the first-order diffracted beam fibre-coupled into the confocal setup. Two 550 nm long-pass filters (Thorlabs FEL550) were used to filter off reflected laser light from the collected emission, which was then sent either into an avalanche photodiode (APD) (SPCM-AQRH-14-FC, Excelitas Technologies) for recording photon count traces and observing the intensity of emission, or to a CCD-coupled spectrometer (Acton Spectrograph, Princeton Instruments) via single-mode optical fibres (SM450 and SM600) for photoluminescence spectroscopy measurements. White-light images of the device were collected by introducing the flippable mirror to divert the collection path to a CCD instead of the detection arm. This allowed easy device positioning and identification. Intensity-correlation measurements were carried out using a Hanbury Brown and Twiss interferometry setup using a 50:50 fibre beam-splitter and a time-to-digital converter (quTAU, qutools) with 81-ps resolution.

**Optically detected magnetic resonance measurements**. ODMR measurements were performed on the confocal setup described above. A 20-μm microstrip microwave antenna was patterned photolithography over the hBN layers and thermal evaporation of 100-nm Au on 20-nm Ti. The antenna was bonded to a coplanar waveguide on a printed circuit board (PCB), shorting the waveguide at the device. A 70-Hz square-wave modulation was applied to the microwave amplitude to detect the change in PL counts as a function of microwave frequency. A permanent magnet delivers an external static magnetic field in the plane of the hBN surface and is changed in strength and orientation by displacing the magnet.

## Data availability

The datasets generated as part of the current study are available from the corresponding authors upon reasonable request.

## Code availability

The codes used for the analysis included in the current study are available from the corresponding authors upon reasonable request.

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

## Acknowledgements

The authors would like to thank L. Bassett, A. Alkauskas and S. Stranks for useful discussions. We thank the research group of Sam Stranks for use of their laser for the EPR measurements. This work is funded by ERC Proof-of-Concept grant PEGASOS (862405), ERC Advanced Grant PEDESTAL (884745), Australian Research Council (CE200100010), and the Asian Office of Aerospace Research & Development (FA2386-20-1-4014). H.L.S is funded by Trinity College, Cambridge. S.E.B. acknowledges funding from the EPSRC CDT in Nanoscience and Nanotechnology (NanoDTC, Grant No. EP/S022953/1). Q.G. acknowledges financial support by the China Scholarship Council and the Cambridge Commonwealth, European & International Trust. S.S. acknowledges funding from the ERC Synergy Grant SC2 No. 610115.

## Author contributions

M.A. and H.L.S. conceived the project. H.L.S., Q.G., J.J., S.E.B. and S.S. performed experiments and carried out analysis. N.M., D.C., H.H.T. and I.A. provided the samples. M.A., H.L.S., Q.G., J.J, S.E.B., N.M., S.S., H.S. and I.A. contributed to discussion of the results and the preparation of the manuscript.

## Competing interests

The authors declare no competing interests.
