## [Peer Review File · Nature Communications]

REVIEWER COMMENTS

Reviewer #1 (Remarks to the Author):

In this paper, H. L. Stern et al. demonstrate for the first time optical detection of the magnetic resonance (ODMR) of single carbon-related defects in hexagonal boron nitride (hBN) at room temperature. These spin defects are currently attracting a deep scientific interest for the development of quantum technologies based on two-dimensional materials. The authors show that ODMR can only be observed for a subset of about 5% of the studied defects, with a contrast being either positive or negative. These observations explain the very low ODMR contrast observed in previous works on large ensembles of carbon-related defects in hBN and shed new light on the complex magneto-optic properties of these defects. Besides the results presented in the main text, the authors provides an extensive characterization of their optical and spin properties, with very useful data added in the supplementary information. I believe that this work deserves publication in Nature Communications. The authors should however address the following comments.

1-Could the authors indicate if the ODMR contrast is always positive for ensemble measurements ? I would expect that this contrast should also be found randomly positive or negative in different regions of the sample.

2-A very recent paper (ref. 40) has reported ODMR on single defects in hBN at cryogenic temperature. These defects have also been associated to carbon-based impurities. It would be interesting that the authors explicitly mention this point and discuss if they are studying the same defect.

3-The data shown in Figure 4(e) indicate a modification of the ODMR spectra when the external magnetic field is applied along the c-axis. The authors should explain how they interpret this observation. I do not really understand why the doublet structure should disappear while considering a spin 1 defect.

4-A variation of magneto-optical properties induced by local strain and electric fields was discussed in a recent theoretical paper focused on boron-vacancy defects in hBN (ref. 36 of the manuscript) The authors might refer to this work in the discussion section.

Reviewer #2 (Remarks to the Author):

The authors report the experimental investigation of individual emitters in 2D hBN showing optically detected magnetic resonance. A variety in ODMR responses is presented, including the 5% yield among emitters, a variety in the ODMR sign, an increase to >30% in ODMR contrast in individual defects compared to 0.4% in ensembles, and the effects of in-plane and out-of-plane magnetic field to the energy level splitting. The results are technically sound, timely and will inspire further work in DFT modeling, experimental photonics and spintronics with applications such as quantum sensing and quantum memories.

In the current form of the manuscript, I find that the scope of work and the narrative are not appealing to the broad readership of Nature Communications, and would be more suitable for journals akin to Nano Letters. In terms of scope, the paper could be expanded with either strain effects studies, theoretical modeling, or coherence studies. In terms of narrative, the introduction could be rewritten to reflect the emerging opportunities and challenges with emitters in 2D materials, and the discussion (which alongside abstract is the most compelling part of the paper) could include implications of the observed variations in ODMR toward practical applications in quantum technologies.

Reviewer #3 (Remarks to the Author):

Stern et al. reports on the study of carbon-related defects in hexagonal boron nitride using optically detected magnetic resonance measurements. They observed elaborate ODMR signals from a small percentage of single defects, including high signal contrast, bipolarity, and doublet resonance in lineshape. Some tentative assignment was given to the origin of the doublet resonance lineshape. Overall, this work reports more detailed ODMR signatures of the carbon-defects in hBN compared to their original work performed at the ensemble level (Nat. Mater. 20, 321-328 (2021)). The experiments were carefully carried out and comprehensive, and the manuscript is well written. Although I appreciate the efforts the authors put in building the statistics and providing thorough experimental details, I feel the manuscript in its current form has very limited discussions about the atomic and electronic origins of these results. Before I would recommend its publication in Nature Communications, the authors should consider expanding their discussions on these aspects, potentially by including theoretical calculations, to help deepen current understandings of these defects. Below I list my comments in detail.

- The PL spectra of the single defects have multiple peaks. Please discuss about the origins of these peaks. Were all these spectral features included in the ODMR measurements? Does selective exclusion of one or two of them affect the ODMR signal?

- Is there any correlation between the g_2 value and the observation of ODMR? How likely is it that the simultaneous detection of several defects, which can still give a $g_2 < 0.5$, cancels out each other's ODMR signal?

- The authors observed that the detection of ODMR is correlated with bunching. In many quantum emitters (e.g. NV centers), the bunching effect is typically related to the existence of a long-lived, dark/meta-stable state. If this is indeed the case for these carbon-related defects, it's likely that the detection of ODMR in only 5% of the defects has an electronic structure-related origin. The authors should consider combining their experimental results with theoretical calculations to see if there are any optically inactive states in the band gap.

- For the assignment of the doublet, can the authors provide more angles and/or orientations between the sample and magnet to confirm their speculation?

Reply to Reviewer Comments

Reviewer #1 (Remarks to the Author):

In this paper, H. L. Stern et al. demonstrate for the first time optical detection of the magnetic resonance (ODMR) of single carbon-related defects in hexagonal boron nitride (hBN) at room temperature. These spin defects are currently attracting a deep scientific interest for the development of quantum technologies based on two-dimensional materials. The authors show that ODMR can only be observed for a subset of about 5% of the studied defects, with a contrast being either positive or negative. These observations explain the very low ODMR contrast observed in previous works on large ensembles of carbon-related defects in hBN and shed new light on the complex magneto-optic properties of these defects. Besides the results presented in the main text, the authors provides an extensive characterization of their optical and spin properties, with very useful data added in the supplementary information. I believe that this work deserves publication in Nature Communications. The authors should however address the following comments.

We thank the reviewer for their positive comments and recommendation for publication.

1-Could the authors indicate if the ODMR contrast is always positive for ensemble measurements ? I would expect that this contrast should also be found randomly positive or negative in different regions of the sample.

The ODMR contrast of the ensemble reported in the manuscript was indeed always positive, and we found this to be the case even across different ensemble samples with varying defect densities. The absolute value of the contrast changes across the ensembles sample, likely because we are sampling varying amounts of ODMR-active defects in each measurement.

Our updated manuscript provides a new kinetic model (new Figure 3 and Section 1d in Supplementary) that explains why single defects can show both positive and negative ODMR contrast. The bipolar contrast we observe is due to the variation in the balance of optical rates dictating the photodynamics and spin polarisation of the system. While both negative and positive contrast is equally likely in our experimental data, we note that the positive defects on average show higher contrast (maximum contrast of 35% for a positive contrast defect, Table 3 Supplementary). This may be the reason that the ensemble is always net positive. We have significantly expanded on the discussion of the origin of the ODMR contrast sign in the main text and the SI.

2-A very recent paper (ref. 40) has reported ODMR on single defects in hBN at cryogenic temperature. These defects have also been associated to carbon-based impurities. It would be interesting that the authors explicitly mention this point and discuss if they are studying the same defect.

We thank the reviewer for this suggestion; indeed the question of defect identity is an important one. Unfortunately, we cannot conclusively state whether we are observing the same defect or not. The hBN studied by Chejanovsky *et al.* (Nat. Materials, 2021) (ref 40) is single crystalline, whilst ours is highly defective and grown via MOVPE; with defects with very different photoluminescence properties (ZPL positions of 750 nm versus our 600 nm). Further, the ODMR behaviour that we see shares some similarities (positive or negative contrast), but also some fundamental differences: they are unable to observe ODMR at room temperature from their defects. Further, they do not report any fine structure in their observed resonances, whilst we regularly observe doublets. Unfortunately, as we know that measured ODMR behaviour is highly variable between defects on the same sample, it is difficult to compare their findings (inferred from studying 3 ODMR-active defects) to the trends observed in our measurements (inferred from studying 27 ODMR-active defects). We have added a spin model in our revised version, and hope that future theoretical and experimental work could verify whether these two classes of defect are indeed the same. We have expanded our discussion of ref. 40 in the main text to explicitly address this question, as the reviewer has suggested (bottom of Page 12, lines 343-351, of manuscript).

3-The data shown in Figure 4(e) indicate a modification of the ODMR spectra when the external magnetic field is applied along the c-axis. The authors should explain how they interpret this observation. I do not really understand why the doublet structure should disappear while considering a spin 1 defect.

We agree with the reviewer that the discussion on this should be expanded, and we have added new Figures 4 and 5 (Figure 5 shown below), and more discussion in the Supplementary Information, to present our angular dependent ODMR in more detail. Inspired by the reviewers' comments we have also extended our analysis of the angular dependent behaviour to develop a computational model for how this occurs, and this discussion has now been added to the main text.

The model demonstrates how a $S=1$ defect with small zero-field splitting parameters ($D = 25$ MHz, $E = 5$ MHz) can present variation in the measured splitting, depending on the relative angle between the applied magnetic field and the defect's \mathbf{D} tensor symmetry axis (z') (determined by the dominant eigenvalue of the \mathbf{D} tensor and referred to by reviewer as the c -axis). When the magnetic field in our experiment is applied down z' the splitting is determined by the zero field parameter D . In our model $D = 25$ MHz, thus a splitting of $2D$ can be resolved for a $S=1$ system. However, if the magnetic field is applied orthogonal to z' , the splitting is instead defined by both D and E and can be smaller than $2D$. When the splitting is smaller than of the constituent resonance linewidths, the resonance will appear a singlet.

We have added more discussion on this point to page 10 lines 287-290.

Our model is consistent with all the defects that we have studied. In the new Figure 5 we show results (model prediction vs measured resonance for each defect) for two defects, and the

results for our other defects are in the Supplementary Figures 32-41. and have added this extended analysis to the Supplementary Information for completeness.

Figure 5: Angular dependence of ODMR below saturation. (a) ODMR spectra (circles) for defect A at two magnetic field orientations, along the z axis (red) and in the x-y plane (orange), magnetic field vectors represented in (c). The solid curves are fits to the data using a $S=1$ model with $D = 25$ MHz, $E = 5$ MHz and the defect \mathbf{D} tensor symmetry axis (z') rotated in the plane of hBN. (b) The simulated ODMR contrast for defect A through the red plane shown in panel c. (c) Schematic showing the magnetic field orientations (red and orange vectors) for measurements in (a) and (b). The red vector points along z, and the orange points out of the xy plane. (d) ODMR spectra (circles) for defect E at a series of magnetic field orientations through a plane, shown in panel f. The solid curves are the fit the $S=1$ model with $D = 25$ MHz, $E = 5$ MHz and z' rotated in the hBN plane (e) The simulated ODMR contrast for defect E through the blue plane shown in panel f. (f) Schematic showing the magnetic field orientations (blue vectors) for measurements in (d) and (e). The blue vectors point at different angles in zy plane with a projection onto x. The sample orientation is shown in both (c) and (f) but the cartoon image (in zy plane).

4-A variation of magneto-optical properties induced by local strain and electric fields was discussed in a recent theoretical paper focused on boron-vacancy defects in hBN (ref. 36 of the manuscript) The authors might refer to this work in the discussion section.

We thank the reviewer for this suggestion. In discussion of our observations regarding the kinetic model and the high variability in optical rates and therefore ODMR contrast for each defect, we have added comment that this is likely to be due to strain. We have added a second citation to ref 36 at line 326.

Reviewer #2 (Remarks to the Author):

The authors report the experimental investigation of individual emitters in 2D hBN showing optically detected magnetic resonance. A variety in ODMR responses is presented, including the 5% yield among emitters, a variety in the ODMR sign, an increase to >30% in ODMR contrast in individual defects compared to 0.4% in ensembles, and the effects of in-plane and out-of-plane magnetic field to the energy level splitting. The results are technically sound, timely and will inspire further work in DFT modeling, experimental photonics and spintronics with applications such as quantum sensing and quantum memories.

In the current form of the manuscript, I find that the scope of work and the narrative are not appealing to the broad readership of Nature Communications, and would be more suitable for journals akin to Nano Letters. In terms of scope, the paper could be expanded with either strain effects studies, theoretical modeling, or coherence studies. In terms of narrative, the introduction could be rewritten to reflect the emerging opportunities and challenges with emitters in 2D materials, and the discussion (which alongside abstract is the most compelling part of the paper) could include implications of the observed variations in ODMR toward practical applications in quantum technologies.

We thank the reviewer for their positive comments on the merit of our results and that the work will inspire further theoretical and experimental work. Regarding scope and narrative, we have taken the reviewer's feedback on board and have significantly modified our manuscript, including a new quantitative kinetic and spin model, two brand new figures and an updated introduction and discussion. New text is highlighted in the revised manuscript.

Our kinetic and spin models shed new light on the structure of the defect and open new routes to a highly tuneable spin-photon interface technology. Our new kinetic analysis, that uses data from numerous defects, shows that we statistically identify a third timescale in defects that show ODMR, which is absent in defects that do not show ODMR. We fit the data from our defects to a three-level kinetic model which can account for the variation in ODMR contrast sign and magnitude that we observe. This model indicates that all the defects we measure are a single optical defect, with a high variability in optical rates, possibly from a variation in local strain. We discuss in the main text how this finding opens routes to using strain to control the spin read-out for hBN defects in future applications of a spin-photon interface, such as in sensing, or quantum networks (lines 327-329).

In addition, our updated manuscript provides a quantitative spin model that is consistent with all of our new angular-dependent ODMR measurements. We measure a series of defects at different orientations of the external magnetic field, and find that we observe a variation in ODMR splitting, consistent with a $S > 1/2$ state where the defect symmetry axis is oriented differently relative to the external magnetic field. In particular, we find that a $S=1$ model with $D = 25$ MHz and $E = 5$ MHz, can explain all of our data. We discuss how this result will advance the field by providing a model to identify hBN single spin defects (line 343).

Reviewer #3 (Remarks to the Author):

Stern et al. reports on the study of carbon-related defects in hexagonal boron nitride using optically detected magnetic resonance measurements. They observed elaborate ODMR signals from a small percentage of single defects, including high signal contrast, bipolarity, and doublet resonance in lineshape. Some tentative assignment was given to the origin of the doublet resonance lineshape. Overall, this work reports more detailed ODMR signatures of the carbon-defects in hBN compared to their original work performed at the ensemble level (Nat. Mater. 20, 321-328 (2021). The experiments were carefully carried out and comprehensive, and the manuscript is well written. Although I appreciate the efforts the authors put in building the statistics and providing thorough experimental details, I feel the manuscript in its current form has very limited discussions about the atomic and electronic origins of these results. Before I would recommend its publication in Nature Communications, the authors should consider expanding their discussions on these aspects, potentially by including theoretical calculations, to help deepen current understandings of these defects. Below I list my comments in detail.

We thank the reviewer for their positive comments on the comprehensiveness of our study, and their suggestion to complement our experimental results with a more in-depth discussion of their implications and the inclusion of some calculations. We have developed two computational models (spin and kinetics) (new Figures 3 and 5) to address this further, which has allowed us to present a more informed discussion of the electronic nature of the defects and origin of the magneto-optical response. We have also explicitly addressed the question of the atomic identity of the defects by adding further discussion of computational studies from the literature, as we believe that extensive modelling of potential defects has already been carried out and is outside the scope of our experimental study. We hope that our new manuscript, which proposes an electronic and spin model for the defects studied, can inspire further theoretical and experimental work to conclusively close in on the atomistic identity of these defects.

- The PL spectra of the single defects have multiple peaks. Please discuss about the origins of these peaks. Were all these spectral features included in the ODMR measurements? Does selective exclusion of one or two of them affect the ODMR signal?

The PL spectra of single defects indeed show multiple peaks, corresponding to the zero-phonon line (highest energy peak) and phonon sideband(s) (lower energy peak(s)) of each defect. These spectra are consistent with lineshapes and energies observed for single defects in other reports (Tran et al., *Nat. Nanotechnol.* (2016), Mendelson, *et al. Nat. Mater.* (2021) and refs 29-33 and 38-45 in the main text).

ODMR measurements were not spectrally filtered, instead the entire emission was measured in our experiments. The reason for this was primarily that the ZPL energies vary from defect to defect, such that we would have to reconfigure our filter requirements for each defect.

We note that we did spectrally filter the ensemble, to measure the ODMR response from different regions of the broadened spectrum, but this had no impact on the presence of ODMR and measurements where the PL spectrum was weaker, were harder to acquire.

- Is there any correlation between the g^2 value and the observation of ODMR?

We thank their reviewer for pointing this out; indeed we had some preliminary indications (originally in the Supplementary of the submitted version) that the dynamics underpinning $g^2(t)$ measurements were related to the presence of ODMR. In light of the reviewer's comments we have moved the discussion of the $g^2(t)$ s from the Supplementary to the main text and expanded on it significantly.

To be clear, we do not see correlations between the $g^2(t)$ value at time zero, which represents the antibunching or 'single-photon' nature of the defect, and the observation of ODMR. This serves simply to confirm the presence of a single defect. However, the bunching behaviour, ie value of $g^2(t)$ at longer time delays than 0, gives us interesting information about the defects. We can confirm that we do observe a correlation between the number of bunching timescales present and the presence of ODMR. Discussion and analysis of this finding is now in Figure 3 (shown below) and expanded on in the Supplementary.

Figure 3: Bunching timescales of hBN ODMR active defects. (a) Second order intensity-correlation ($g^2(\tau)$) measurement of an ODMR-active defect, defect B, measured at $1.5P_{\text{sat}}^{\text{optical}}$ excitation (laser power saturation in Supplementary Figure 3), showing the dynamics out to 1-ms time delay, non-background corrected. The data (circles) is fit to a bi- and tri-exponential fit (solid curves). (b) $g^2(\tau)$ measurement of a defect that does not show ODMR, out to 1-ms delay, measured at $0.2P_{\text{sat}}^{\text{optical}}$ excitation, non-background corrected. For background correction analysis see Supplementary Fig. 11. The grey circles are the data and the solid line is a bi-exponential fit. (d,e,f) The distribution of antibunching (τ_{ab}) and bunching (τ_{b} and $\tau_{\text{b(additional)}}$) timescales from 40 measurements of 18 defects. Data for defects that show ODMR is in red and defects that do not show ODMR in blue. Defects that don't show ODMR are not plotted in (e) because this data contains high error, as shown in (c). (c) A scatter plot (left plot) and histogram (right plot) of the error on the fractional error on the fit ($\sigma_{\text{b(add)}/\tau_{\text{b(additional)}}$) of the additional bunching timescale, for ODMR and non ODMR active defects.

In brief, the presence of two bunching timescales for ODMR-active defects is explained by our three-level kinetic model that we present in the updated manuscript. This model (Supplementary section 1d) shows how a range in shelving (k_{23} and k_{53} in model below) and de-shelving (k_{34} and k_{31}) optical rates across the defects results in tuning the ODMR contrast magnitude and sign, for a given optical defect. Our model is consistent with the laser-power dependent $g^2(t)$ measurements that we have collected for ~ 20 defects (half ODMR active, half ODMR inactive). We suggest in the text that this high variability in optical rates for the hBN defects is related to strain in the material, opening routes towards strain control of the spin readout.

LEFT: Figure 13 replicated from the Supplementary showing the kinetic model that is consistent with our defects. RIGHT: Figure 18 from Supplementary showing the ODMR contrast response when the shelving and de-shelving rates are tuned.

How likely is it that the simultaneous detection of several defects, which can still give a $g^2 < 0.5$, cancels out each other's ODMR signal?

Theoretically, for a single photon emitter, $g^{(2)}(0) = 0$. However in experiments, $g^{(2)}(0)$ for single-photon emitters are typically not equal to 0 due to experimental factors but the threshold of $g^{(2)}(0) < 0.5$ is typically used to indicate single photon emission.^{1,2} From our $g^2(\tau)$ analysis we show that the defects we measure show $g^{(2)}(0) < 0.5$ and therefore are likely to be single defects not several defects.

We have included extra analysis in the Supplementary on the effect of this 'noise' on the value of $g^2(0)$ (Figure 11, Supplementary). This analysis shows that the detector jitter associated with our detector's time resolution, and the use of finite bin widths in the processing of our data will increase the value of $g^2(0)$, meaning our quoted values are larger than the true value of $g^2(0)$ which we cannot experimentally access^{3,4,5,6}.

While it may possible that some of our defects are double defects (with one defect being much dimmer than the other), we don't see any clear evidence for this in other measurements (eg. multiple timescales in time-resolved PL or multiple ZPLs in PL lineshape). Further, we don't

regard the quality of the single photon emission as the main message of paper- rather the identification of an hBN spin defect at room temperature.

- *The authors observed that the detection of ODMR is correlated with bunching. In many quantum emitters (e.g. NV centers), the bunching effect is typically related to the existence of a long-lived, dark/meta-stable state. If this is indeed the case for these carbon-related defects, it's likely that the detection of ODMR in only 5% of the defects has an electronic structure-related origin. The authors should consider combining their experimental results with theoretical calculations to see if there are any optically inactive states in the band gap.*

We agree with the reviewer that the bunching effect is consistent with the presence of a metastable state and that this required further investigation. As mentioned earlier in our response, we have developed a kinetic model and calculated how the presence of such a state could be consistent with our experimental results. Indeed, the kinetic model strongly supports the hypothesis that the detection of ODMR contrast is related the balance of the rates that link the metastable state to the optical manifold. This is explained extensively in the Supplementary information. As such, our model indicates the ODMR yield is related to the variation in optical rates for each defect rather than the presence of an additional electronic state, although this is in principle a possibility.

- *For the assignment of the doublet, can the authors provide more angles and/or orientations between the sample and magnet to confirm their speculation?*

We thank the reviewer for this suggestion, which inspired further experiments and the addition of the angular-dependent spin model discussed previously (new Figure 5 and Supplementary Figures 32-41). Essentially, the model demonstrates how a S=1 defect with small zero-field splitting parameters can present doublet or singlet character depending on the relative angle between the applied magnetic field and defect symmetry axis. Again, we hope that these results will enable further theoretical work into identifying which atomic structures might be consistent with the models we have developed and the statistics we have observed.

References

- 1 Fox, Mark. *Quantum optics: an introduction*. Vol. 15. OUP Oxford, 2006.
- 2 Eisaman *et al.*, Single-photon sources and detectors. *Rev. Sci. Instrum.* 82, 071101 (2011).
- 3 Bommer, A., and Becher, C. New insights into nonclassical light emission from defects in multi-layer hexagonal boron nitride. *Nanophotonics*, **8**, 11, 2041–2048, (2019).
- 4 Thorn, JJ., *et al.*,. Observing the quantum behavior of light in an undergraduate laboratory. *American Journal of Physics*, **72**, 9, 1210–1219, (2004).
- 5 Berthel, M., *et al.*, Photophysics of single nitrogen-vacancy centers in diamond nanocrystals. *Physical Review B*, **91**, 3, 035308, (2015).
- 6 Exarhos, A.L. *et al.* Magnetic-field-dependent quantum emission in hexagonal boron nitride at room temperature. *Nat. Commun.* **10**, 222 (2019).

REVIEWERS' COMMENTS

Reviewer #1 (Remarks to the Author):

The authors have responded well to all my questions and comments. I think the paper has been significantly improved. It deserves to be published in Nature Communications.

Reviewer #2 (Remarks to the Author):

The authors have significantly expanded the study of the origin of ODMR signal of the emitters in 2D hBN. Their analysis shows the intriguing relationship between the bunching dynamics and the occurrence and the sign of the ODMR. I find the paper in its current form to be highly relevant for the readership of Nature Communications.

Reviewer #3 (Remarks to the Author):

The authors have addressed my comments thoroughly. I would recommend the publication of their manuscript in Nature Communications as it is.